# Displacement, personal loss, and psychological strain among physicians and nurses working in Gaza, 2023–2024

Mohanad S. A. Qudaih[1], Hamza A. Abu Daqqa[2], Omar R. AlNajjar[2], Dalia T. A. Wehedi[3], Rasha Khoury[4], Benjamin Bouquet[5], Lisa Matos[6], Karim Sariahmed[7]*

1 Faculty of Medicine, Al-Quds University, Jerusalem, Palestine, 2 Faculty of Medicine, Islamic University, Gaza, Palestine, 3 World Health Organization – Gaza Office, Gaza, Palestine, 4 Department of Obstetrics & Gynaecology, Division of Fetal Medicine and Complex Family Planning, Boston University School of Medicine, Boston, Massachusetts, United States of America, 5 Population Health Sciences Institute and School of Law at Newcastle University, Newcastle, United Kingdom, 6 Instituto de Ciências Psicopedagógicas - Instituto Universitário, Lisbon, Portugal, 7 Section of General Internal Medicine, Department of Medicine, Boston University School of Medicine, Boston, Massachusetts, United States of America

* karim.sariahmed@bmc.org

## Abstract

On January 26th 2024, the International Court of Justice (ICJ) recognized plausible grounds for genocide being committed in Gaza by Israel. A hallmark of the violence has been unprecedented attacks on health workers since October 7th, 2023. We use the word "genocide" to refer to this period of accelerated violence and displacement in Gaza, following ICJ findings, detailed reports by human rights organizations, and statements by genocide scholars concluding that Israel is indeed committing genocide. To assess impacts of this violence, between December 2023 and January 2024 we conducted an anonymous cross-sectional online survey of physicians and nurses working in Gaza prior to and during the genocide. Participants reported on workplace displacement, personal losses, and psychological impacts using the Professional Quality of Life Scale for Health Workers (ProQOL-H). 56 participants completed one or more survey components. Among 46 participants reporting on displacement, 25 (54%) reported having to change workplaces at least once. Among 41 participants documenting personal loss, 17 (41%) reported that a family member was killed and 32 (78%) reported that a close colleague was killed. Among 35 participants completing the ProQOL-H, 17 (49%) respondents reported "high" compassion satisfaction (scores of 24–30), 13 (37%) reported high burnout, and 13 (37%) reported high secondary traumatic stress. This is markedly different from CS, BO, and STS scores reported by nurses in the West Bank in a recent study, likely reflecting the genocide in Gaza. Further work with survivors is needed to characterize their experiences and reconcile them with culture-specific coping strategies such as *sumud.* Culturally relevant, mixed methods follow up to this work is needed to inform interventions to

**Data availability statement:** Data are available in the manuscript or accompanying supporting information.

**Funding:** The author(s) received no specific funding for this work.

**Competing interests:** The authors have declared that no competing interests exist.

support the recovery of survivors and rebuild Gaza's healthcare system. This will only be possible when the genocide is brought to a permanent end, along with the conditions which have enabled it.

## Background

Israel has a settler-colonial relationship with Palestine and Palestinians, which means that, unlike other colonial regimes focused on exploitation or enslavement, the chief goal is to eliminate the indigenous Palestinians [1,2]. This is partly realized through Israel's related and widely condemned crime of apartheid, a system of colonial race domination codified in Israeli law which deprives Palestinian people of freedoms and entitlements that are taken for granted as basic human rights in other contexts [1]. Daily life under this regime resembles the Jim Crow South. For example, hundreds of thousands of Palestinians in the West Bank and Gaza have been arrested since 1967 and tried in military courts which fail to meet international standards of fair trial, while Jewish settlers living in illegal settlements on occupied territory in the West Bank are exempt from the military orders governing Palestinians [3]. On January 26th 2024, the International Court of Justice (ICJ) recognized plausible grounds for genocide being committed in Gaza by Israel [4]. In the report "Anatomy of a Genocide," the United Nations Special Rapporteur on Palestinian territories occupied since 1967 Francesca Albanese presents the evidence that Israel's assault on Palestinians in Gaza constitutes genocide, situating these findings in relation to the eliminatory context of settler-colonialism that affects the entirety of occupied Palestine [5]. Evidence of genocide presented in the report included details of the following acts: killing members of the group, causing serious bodily or mental harm to groups' members, and deliberately inflicting on the group conditions of life calculated to bring about its physical destruction in whole or in part, including the destruction of healthcare, food, and sanitation systems. Albanese also presented evidence that these acts were carried out following repeated statements of genocidal intent made by senior Israeli military and government figures. In the words of the Special Rapporteur, "When genocidal intent is so conspicuous, so ostentatious as it is in Gaza, we cannot avert our eyes, we must confront genocide."[6].

The targeting of the healthcare system and health workers in Gaza has been a particularly egregious feature of the recent assault. The aim of our study is to learn about the violations suffered by health workers in Gaza during this period, as measured by their loss of loved ones and colleagues and their workplace displacement, and its psychological consequences, and measured by the ProQOL-H, which is described in detail below. To interpret the data in the context of genocide, we draw on literature on severe trauma and its consequences, existing literature on culturally specific strategies for coping with trauma in Palestine, and the broader literature on the well-being of health workers and its importance for health system functioning.

## Targeted attacks on healthcare in Gaza and their impact on health workers

The Gaza Strip has endured repeated attacks in recent years resulting in tens of thousands of deaths and injuries, repeated mass population displacement [7], and the widespread destruction of health services and public health infrastructure [8]. Repeated attacks on health workers in Palestine have been documented across many years of Israeli military attacks on Palestine [9–11]. Since October 7th 2023, there has been an unprecedented increase in the number and severity of Israeli attacks on the health system in Gaza, resulting in the death and injury of hundreds of health workers, patients and caretakers as well as damage to essential health infrastructure [12]. The health system has all but collapsed, with the World Health Organization (WHO) reporting that as of February 28th, 2025, 886 people have been killed in attacks on healthcare since October 2023, while only 18 hospitals in Gaza are partially functional, leaving the remaining 20 hospitals completely out of service. The Palestinian Ministry of Health reported that 986 health workers were killed as of September 2024 [13]. Within just the first six weeks of the assault, 25% of hospitals in Gaza had craters left by 2000-lb bombs within lethal range and 83.3% had these craters within infrastructure damage and injury range [14], and spatial analysis confirms that the destruction of civilian infrastructure including health systems is not random or accidental [15]. This is further supported by documentation of the hundreds of health workers abducted and often tortured or killed in Israeli prisons [16–18]. Weeks after the withdrawal of Israeli occupying forces from Al-Shifa Hospital following a horrific ground invasion, Palestinian health workers uncovered a mass grave near the hospital containing the remains of patients, some with wound dressings and catheters still in place [19]. Shortly thereafter, an even larger mass grave was discovered near Nasser Hospital, with some bodies found with bound hands or feet [20]. While not the first instance of attacks on health workers in Palestine, it is unprecedented in its brutality. In November 2023, a Médecins Sans Frontières (MSF) nurse texted his colleagues from the basement of Al-Shifa Hospital, where he and his family were taking shelter during intense Israeli bombardment: "We are being killed here, please do something."[21].

## Mental health in Palestine

Many prior studies have measured the psychological burdens carried by healthcare workers in Palestine related to the baseline state of working while subject to Israeli settler-colonialism. Abu-el-Noor et al found that approximately 90% of doctors and nurses surveyed likely had probable posttraumatic stress disorder (PTSD) following Israeli attacks in Gaza in 2014 [22]. A follow up study two years later showed minimal changes in reported symptoms, which they attributed to minimal interventions [23]. Veronese et al measured posttraumatic growth and sense of coherence, two related constructs which focus on psychological resources for making meaning out of suffering and mobilizing individual-level coping strategies [24]. Recognizing the many threats to validity of PTSD symptom checklists and other self-reporting instruments as well as debate over the validity of PTSD in the Palestinian context [25], Afana et al described culture-specific expressions for both coping resources and psychological distress, many of which frame Palestinian suffering as it relates to their national cause [26]. Particular attention has been given to the notion of "sumud", translated as steadfastness, which is profoundly important in Palestinian society [27]. Wick characterizes sumud as a coping mechanism used by Palestinian health workers to navigate the daily challenges associated with their jobs [28]. While sumud is indeed an intergenerational mental health resource, it is also fundamentally a cultural and psychological tool to resist colonial oppression [29]. As Meari describes, sumud is a practice that allows Palestinian political prisoners to be both victims of colonial oppression and agents who resist that oppression at all costs. The concept of sumud, rooted in the forced displacement of Palestinians since the Nakba of 1947–9, takes on new meaning in the context of the ongoing, accelerated genocide [30–33].

## Psychological constructs for health worker distress and well-being

Reflecting health workers' frequent exposure to trauma [34–36] and its consequences for health system functioning [34], the broader psychological and health services literatures on the distress and well-being of health workers is vast. Many

related and sometimes overlapping concepts have been developed to describe it, including vicarious traumatization [36], compassion fatigue, compassion satisfaction, secondary traumatic stress, burnout, perceived support [37], and moral distress [38]. While there is variation in the definition and application of these concepts, vicarious traumatization generally refers to the stress response of professions with an ongoing empathic relationship with a traumatized client, while secondary traumatic stress (STS) typically refers to an acute stress response which can be experienced by many kinds of health workers when exposed to the suffering of others [36,39]. Burnout (BO), like STS, is applied to many kinds of health workers but reflects a more gradually lost sense of accomplishment and achievement and an erosion of idealism [36,39]. Compassion fatigue (CF) and its inverse, compassion satisfaction (CS) reflect a combination of STS and BO [40], and have also been used to describe the experiences of lay caregivers [39]. Moral distress (MD) is the perception that one's ethics are compromised due to aspects of their work environment which they cannot control [38]. All of these constructs have been found to play some role in the health worker's ability to cope with work-related stress. The Professional Quality of Life Scale (ProQOL) incorporates several of these concepts, and was recently administered to Palestinian ICU nurses in the West Bank, with 80% or more reporting average levels of compassion satisfaction, burnout, and secondary traumatic stress [41]. The study team consulted nursing and psychometric experts to approve the tool's cultural and contextual relevance and construct validity, but did not explicitly explore the political context and its potential bearing on the findings.

**Studying the psychological effects of genocide**

While existing literature acts as a guide to the study of mental health in Palestine generally and of health workers in Palestine, the current situation for health workers in Palestine is uniquely horrific. For those who survive, they continue to live in perpetual proximity to death [42]. The mental health of health workers is important to the functioning of any health system, but the psychological toll of the ongoing genocide must be studied and understood within its violent political context and the targeting of health workers. While the study of nurses in the West Bank in 2024 reflected an average professional quality of life, the brutal treatment of health workers during the genocide in Gaza needs additional attention.

Genocide is qualitatively distinct as a traumatic exposure and our study of it must take this into account [2]. Alongside the destruction of the healthcare system in the determination of genocide there is also evidence of the destruction of cultural heritage sites, mosques, and churches, which form an important part of the Palestinian social fabric. These components are important in relation to the study of trauma as the nature of traumatic exposure and its context is known to shape psychological sequelae [43–45]. In the words of Albanese: "Israeli soldiers have published footage boasting about their killing of families, mothers, children - the bombing of homes, mosques, and schools. Self-incriminating videos show them sadistically mocking and humiliating their Palestinian victims not only by violating their physical integrity and right to life, but also their dignity, their most intimate possessions and spaces that the soldiers have entered and looted, and by desecrating cemeteries and places of worship."[5]. Among survivors of the genocide in Rwanda, the peritraumatic response was found to mediate the relationship between the intensity of the traumatic exposure and symptoms of traumatic grief [46]. Their study highlights the importance of measuring psychological experiences proximal to the traumatic exposure and the need to study the specific consequences of genocide as a traumatic exposure, whose nature is highlighted in Albanese's description.

The psychological effects of any form of extreme trauma on oppressed peoples must take into account the social and political context that shape values, beliefs, and life goals [47]. Health workers attacked by the Israeli military during the 2018 Great March of Return, a historic nonviolent uprising in Gaza, pointed to the structural and continuous nature of violence as key features of its impact [48].

In order to assess and contextualize the psychological impact of the ongoing genocide on physicians and nurses working in Gaza after October 7th, 2023, we conducted a brief online survey to assess experiences of workplace displacement, personal loss, and the impact of traumatic stress.

We use the word "genocide" to refer to this period of accelerated violence and displacement in Gaza, following pre-liminary ICJ findings and subsequent detailed reporting by UN Special Rapporteur Albanese [4,5]. It is also consistent with more recent reports of the United Nations Special Committee to Investigate Israeli Practices [49], Human Rights Watch [50], and Amnesty International [51] all finding that Israel has indeed committed genocide in Gaza. These finds are echoed increasingly by genocide scholars [52]. The psychological consequences for health staff enduring this can be understood in relation to past work on the mental health of Palestinian health workers, the mental health of other victims of genocide [46], and the broader literature on the well-being of health workers.

## Methods

### Ethics statement

This research was conducted in accordance with the U.S. Federal Policy for the Protection of Human Subjects (the Common Rule). Boston Medical Center and Boston University Medical Campus Institutional Review Board #: H-44523.

### Study design, setting and population

We conducted an anonymous, English language cross-sectional online survey between December 24, 2023 and January 30, 2024. Participants had to be physicians or nurses that worked as employees or volunteers in Gaza before October 7th, 2023 and continued to work there for at least some of the period following. We chose to focus on a more homoge-nous group of health workers to improve validity since we were expecting a small sample, and we chose to focus on phy-sicians and nurses since they are the majority of frontline health workers in Palestine [53,54]. Both those born in Palestine and their colleagues born abroad but working in Gaza were included. Medical and nursing students were not included. Participants had to be age 18 or older.

### Recruitment

A physician study leader born in Gaza and trained in Palestine led the development of outreach plans, consulting with Palestinian physicians and researchers locally and internationally to determine contexts and methods for outreach which were effective and sensitive. We used the respondent-driven sampling method as described by Wejnert and Heckathorn [55], with which we sought "seed" participants affiliated with institutions in all five governorates of the Gaza Strip. We prompted these seed participants by email to take they survey if they met inclusion criteria, and to disseminate it using a recruitment message to their professional networks, including to colleagues linked to hospitals in Gaza, and Ministry of Health (MOH), UNRWA, and other primary healthcare centers. Additionally, we encouraged participants to share the survey link and outreach message across various channels such as email, health worker group chats, and community networks. It was not shared via other social media.

Of the 26 potential "seed" participants initially invited to complete the survey, four (including the recruitment lead) recruited additional participants. They were situated in North Gaza and Khan Younis before October 7th, 2023. Phys-ical exhaustion, displacement, forced hospital evacuations, attacks on hospitals, and prolonged disruption to internet connectivity hindered survey participation. One co-author's displacement resulted in the loss of their laptop, further impeding study recruitment. Most survey participants were recruited by invitation either in person or via bilateral personal conversations.

Participants provided informed consent on the first page of the online survey. The Boston Medical Center and Bos-ton University Medical Campus Institutional Review Board determined that this study was exempt from full review on the basis of its anonymity and the limited in-person interactions required for participation. Sensitive questions, such as those pertaining to personal loss or psychological distress, could be skipped at the respondents' discretion to mitigate the potential harm caused by active retrieval of traumatic memories. The consent process emphasized the

voluntary nature of the survey and the option to close the browser and end the survey at any time and without need for justification.

### Survey development

We developed and distributed the survey using the web-based Qualtrics Research Suite software. In addition to demographic data, the survey included questions about workplace displacement, personal experiences of death and injury among family members and colleagues, and an instrument for measuring the psychological impact of providing medical care in the context of the genocide (S1 Text). We selected the Professional Quality of Life Scale for Health Workers (ProQOL-H) developed by the Center for Victims of Torture [56]. It assesses compassion satisfaction, burnout, compassion fatigue, moral distress, and secondary traumatization among health workers and other professionals working with victims of torture and other extreme trauma exposures. Questions are answered using a five-point Likert scale. Among our colleagues, we asked those who would potentially qualify as participants for feedback on the draft survey prior to seeking IRB review. We initially intended to use the Arabic version of the survey, however questions 13 and 18 in the Arabic version of ProQOL-H translate "well-being" as *"rafahia."* We felt this translation gave these questions a more frivolous connotation with potential to offend participants, given the context. Creators of the instrument require permission to modify the survey, which we were unable to obtain, so we proceeded with the English language version, without modification. English is spoken by the majority of physicians and nurses in Palestine. The ProQOL, another version of the instrument, has also been administered in English to ICU nurses in the West Bank [41].

Participant demographics, their experiences of displacement and personal loss, and their ProQOL-H scores are presented in the form of simple descriptive statistics. To further explore the ProQOL-H outcomes and generate hypotheses to drive future work, we stratified ProQOL-H outcomes by gender, by profession, by most recent workplace governorate, and by the type of facility lived in.

## Results

### Recruitment

57 surveys were returned. One response was removed as it was suspected to be a duplicate due to substantial similarity with another response in completion time, demographics, and the formulation of open-ended responses. Completion rates varied across survey sections, with 50 (89%) of respondents reporting demographic information, 46 (82%) reporting on workplace displacement, 41 (73%) reporting on personal experiences with death and injury, and 35 (63%) completing the ProQOL-Health instrument. All participants who started the ProQOL-Health completed all 30 questions.

Male health workers and physicians were more represented among participants than female health workers or nurses (Table 1). This gendered and disciplinary skew in recruitment was recognized early and, although physicians in Gaza are disproportionately male [54], recruiters encouraged additional dissemination of the survey among women and nurses. 90% of participants were born Gaza, with the remainder being born outside Palestine.

### Workplace displacement and personal loss

46 respondents reported whether they relocated to new workplaces for various reasons related to the genocide (Table 2, Fig 1). Prior to October 7th 2023 most respondents worked either in Khan Younis (n = 22, n = 14 at European Gaza Hospital) or Gaza City (n = 14, n = 9 at Al-Shifa Hospital). A smaller number of participants were working in Rafah prior to October 7th (n = 6, n = 5 at Kuwaiti Hospital). Many respondents had relocated to work in Rafah (n = 19 of 46 working in Rafah at the time of survey completion, n = 17 at Kuwaiti Hospital).

Personal losses were experienced by 41 health workers as a result of Israeli military violence (Table 3). Among them, 17 of 41 (41%) reported that a family member was killed, and 32 of 41 (78%) reported that a close colleague had been killed. 19 of 41 (46%) reported both a personal loss and workplace displacement.

**Table 1. Survey Participant Demographics, n = 50.**

| Profession | n(%) | Current Homestay - Governorate | n(%) |
|---|---|---|---|
| Physician | 38 (76) | North Gaza | 0 (0) |
| Nurse | 12 (24) | Gaza City | 2 (4) |
| Total | 50 | | |
| **Age** | | Middle Gaza | 8 (16) |
| 18-24 | 7 (14) | Khan Younis | 9 (18) |
| 25-34 | 30 (60) | Rafah | 31 (62) |
| | | Total | 50 |
| 35-44 | 7 (14) | **Current Homestay - Facility** | |
| 45-54 | 5 (10) | With family or friends | 15 (30) |
| 55-64 | 1 (2) | I do not leave the hospital | 14 (28) |
| 65 and older | 0 (0) | At home (not displaced) | 9 (18) |
| Total | 50 | | |
| **Birthplace** | | School | 6 (12) |
| North Gaza | 2 (4) | Refugee camp | 5 (10) |
| Gaza City | 15 (30) | Other | 1 (2) |
| Middle Gaza | 0 (0) | UNRWA shelter | 0 (0) |
| Khan Younis | 19 (38) | Mosque or church | 0 (0) |
| | | Total | 50 |
| Rafah | 9 (18) | **Gender** | |
| West Bank and Jerusalem | 0 (0) | Male | 35 (70) |
| Born outside Palestine | 5 (10) | Female | 15 (30) |
| Total | 50 | | |
| | | Other or Prefer not to say | 0 (0) |
| | | Total | 50 |

## The psychological effects of the genocide on health workers

The quality of responses to the ProQOL-Health scale appeared satisfactory, insofar as there were no instances of "straightlining" (selecting the same response for every item in the grid). Response counts for individual survey items are reported in S1 Table. There was a higher prevalence of "high" compassion satisfaction (scores of 24–30, with 30 being the highest possible score) compared to other subscores, with 49% (n = 17) of respondents reporting high scores (Table 4). 37% (n = 13) of respondents reported high levels of burnout, and 37% (n = 13) reported high secondary traumatic stress. 86% (n = 30) of respondents reported a level of perceived support in the "average" range (scores of 13–23), and 91% (n = 32) of respondents reported moral distress in the average range.

There was significant concordance among most respondents for four questions (S1 Table). Specifically, 27 of 35 respondents reported feeling supported by their families in their healthcare work, of which 18 selected "very often." Similarly, 28 of 35 respondents expressed feeling pride in their ability to help patients, of which 19 selected "very often." There was strong consensus among workers that their workload seems endless (25 of 35 marked "often" or "very often") and that work has significantly impacted their work-life balance (29 of 35 marked "often or "very often").

Stratified analyses did not suggest major differences in ProQOL-H scores based on gender or profession. (S2 Table) Respondents living with family or friends (n = 8) appeared to have lower compassion satisfaction and perceived support than those still at home, staying in the hospital, or living in a refugee camp or school. The single individual working most recently in North Gaza at the time of the survey reported far lower compassion satisfaction and perceived support than the median values reported from those working in Gaza City, Khan Younis, Middle Gaza, and Rafah.

**Table 2. Survey Participant Workplace Displacement, n = 46.**

| Workplace Prior to October 7th, 2023 Governorate | n | Current Workplace - Governorate | n |
|---|---|---|---|
| North Gaza | 4 | North Gaza | 3 |
| Gaza City | 14 | Gaza City | 7 |
| Middle Gaza | 0 | Middle Gaza | 2 |
| Khan-Younis | 22 | Khan-Younis | 15 |
| Rafah | 6 | Rafah | 19 |
| **Workplace Prior to October 7th, 2023 Facility** | | **Current Workplace - Facility** | |
| Al-Ahli Arab Hospital | 1 | Al-Ahli Arab Hospital | 0 |
| Al-Awda Hospital | 0 | Al-Awda Hospital | 1 |
| Al-Shifa Hospital | 9 | Al-Shifa Hospital | 6 |
| Beit Hanoun Hospital | 1 | Beit Hanoun Hospital | 1 |
| European Gaza Hospital | 14 | European Gaza Hospital | 10 |
| Indonesian Hospital | 2 | Indonesian Hospital | 1 |
| Nasser Medical Complex | 6 | Nasser Medical Complex | 5 |
| Algerian Military Hospital | 1 | Algerian Military Hospital | 0 |
| Al Naser Hospital for Pediatrics | 1 | Al Naser Hospital for Pediatrics | 0 |
| Al-Sadaqa Turkish-Palestinian Hospital | 1 | Al-Sadaqa Turkish-Palestinian Hospital | 0 |
| Kuwaiti Hospital | 5 | Kuwaiti Hospital | 17 |
| Al-Helal Emirati Hospital | 1 | Al-Helal Emirati Hospital | 0 |
| An UNRWA Primary Care Center | 1 | An UNRWA Primary Care Center | 2 |
| A MOH Primary Care Center | 1 | A MOH Primary Care Center | 1 |
| Other Facility Not Listed | 2 | Other Facility Not Listed | 2 |

46 health workers reported on displacement, of which 25 indicated that they had been displaced from their workplace at least once. Among 10 respondents who stopped working entirely, reasons included hospital and clinic closures (n = 2), forced evacuations of hospitals (n = 2) and neighborhoods (n = 3), injury (n = 1), or other safety concerns (n = 2).

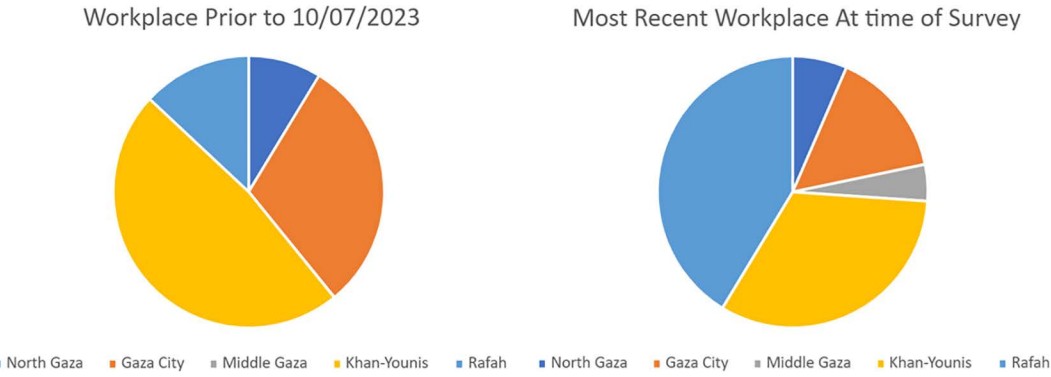

**Fig 1. Workplace displacement reported by doctors and nurses by Gaza Governorate, December 2023 to January 2024 (n = 46).**

Some participants used the open response spaces to describe the challenges and atrocities they encountered and how they have adapted, or to share other aspects of their experiences. These responses were too few to allow for qualitative analysis, but have been compiled in S2 Text. We include one such response from a young physician:

**Table 3. Survey Participants Reporting Personal Loss after October 7th, 2023, n = 41.**

| | n (%) |
|---|---|
| Family member(s) killed in aerial bombardment or the ground invasion | 17 (41%) |
| Family member(s) seriously injured or disabled in aerial bombardment or the ground invasion | 20 (49%) |
| Colleague(s) killed in aerial bombardment or the ground invasion | 32 (78%) |
| Colleague(s) seriously injured or disabled in aerial bombardment or the ground invasion | 31 (76%) |

**Table 4. ProQOL-Health sub-scores with median scores (IQR) and number of scores by category as defined by the Center for Victims of Torture (n = 35).**

| Sub-Score | Median Score (IQR) | Score Category | n (%) |
|---|---|---|---|
| Compassion Satisfaction | 23 (20-25) | High (≥24) | 17 (49%) |
| | | Average (13–23) | 17 (49%) |
| | | Low (≤12) | 1 (2.8%) |
| Perceived Support | 20 (17-22) | High (≥24) | 5 (14%) |
| | | Average (13–23) | 30 (86%) |
| | | Low (≤12) | 0 (0%) |
| Burnout | 21 (20-24) | High (≥24) | 13 (37%) |
| | | Average (13–23) | 22 (63%) |
| | | Low (≤12) | 0 (0%) |
| Secondary Traumatic Stress | 21 (19-25) | High (≥24) | 13 (37%) |
| | | Average (13–23) | 21 (60%) |
| | | Low (≤12) | 1 (2.9%) |
| Moral Distress | 19 (16-20) | High (≥24) | 2 (5.7%) |
| | | Average (13–23) | 32 (91%) |
| | | Low (≤12) | 1 (2.8%) |

*"I did not expect to live this stage or to see the amount of victims and injured like what I saw. I was choosing between patients who needed a quick operation, who could endure another time, and for whom we could do nothing. I refused to leave the hospital [where] I was staying. We continued with the treatment, but we reached the point where we were forced to leave our workplace by force of arms and fire. They arrested my colleagues and killed others. I saw death and I thought that doctors were protected."*

## Discussion

In our study of doctors and nurses working during Israel's genocide against the Palestinian people in Gaza between late December 2023 and January 2024, most respondents reported that they had been displaced from their workplaces. Their experiences of displacement are shared by 1.7 million Palestinians who have been displaced into the southern Rafah governorate [12]. The repeated displacement into increasingly crowded homestays and workplaces is a condition health workers shared with all people in Gaza, and this is a factor in their reported distress and their responses to that distress.

Most participants also reported that close colleagues had been killed, while almost half reported that family members had been killed by the Israeli military. These findings point to the profoundly devastating toll of the genocide on our Palestinian health worker colleagues. We expect the proportion of health workers reporting traumatic loss and direct experience of violence and displacement to be far higher now; Israel has systematically targeted the health sector in Gaza as part of

an effort to render the region uninhabitable, and violence throughout Gaza has continued since the time of the survey [12], including deaths due to Israeli violations of ceasefire terms [57].

ProQOL-H scores among participants revealed high rates of burnout and secondary traumatic stress alongside very high levels of compassion satisfaction and moderate levels of moral distress. Our analysis of compassion satisfaction, burnout, and secondary traumatic stress benefits greatly from the work of Ayed et al with 162 ICU nurses in the West Bank, where they used ProQOL-5, another version of the instrument used here which measures the same 3 constructs, also administered in English [41]. 37% (13 of 35) of health workers presently surveyed reported high levels of burnout compared to 0% (0 of 162) of West Bank ICU nurses. Secondary traumatic stress was also higher in the present study, with 37% (13 of 35) reporting high levels compared to 0.6% (1 out of 162) of West Bank ICU nurses. Doctors and nurses working during the genocide in Gaza also reported very high levels of compassion satisfaction, 49% (17 of 35) reported high levels compared to 4.9% of West Bank ICU nurses (8 of 162). While nurses in the West Bank suffer from the same regime of apartheid and settler colonialism as health workers in Gaza, the stark difference in reported mental health reflects the reality that the genocidal violence in Gaza is unprecedented, despite many prior assaults by Israeli forces which have also affected health workers.

For the categories of perceived support and moral distress, existing literature does not have such a faithful basis for comparison. In both of these categories, the vast majority reported scores in an average range. However, for all categories can be made more meaningful with the context provided in the literature on Palestinian or Arab idioms of distress and coping strategies [26]. While a mixed-methods study is needed to empirically compare the five ProQOL-H concepts to the idioms of distress or coping resources specific to Palestine, existing literature offers context for the high levels of reported compassion satisfaction. In Arab culture generally, trauma is not necessarily understood as an individual experience, and even the concept of selfhood has a more relational character than in Western societies [58]. Afana et al describe positive religious coping and Palestinian understandings of their liberation struggle as reflections of coping as a structural feature of their social network, rather than merely a quality of individuals. Connection to this structure and the view that exposure to trauma is part of the struggle for national liberation may explain the higher level of compassion satisfaction reported by health workers, in Gaza who performed their duties while enduring the existential threat of genocide. These findings and background support the two conceptions of *sumud* by Wick and Meari – the structured capacity to cope with oppression is also a foundation for anti-colonial resistance [28,29]. Still, we must interpret these data with caution. Because perceived support and moral distress were not included in the version of ProQOL used by Ayed et al, the interpretation of these two measures does not benefit from comparison to another sample of Palestinian health workers. Interviews and qualitative analysis would be needed to understand if there is meaningful overlap between ProlQOL-H constructs and *sumud* or other culturally-specific responses to hardship.

Our study has several limitations. Due to the many more pressing needs facing health workers in Gaza and severe limitations imposed by Israel that have affected internet connectivity, our sample is small. The sample also may not be representative of all health workers in Gaza. Some potentially eligible participants told one recruiter that they did not want to share personal information of any kind because of security concerns and the constant fear of being targeted by the Israeli military. This fear underscores the psychological dimension of the assault. Others simply did not think research was a good use of time when facing such a dire situation. We acquired approval for the use of paper-based surveys however these were not ultimately used, since they require more time to process and did not alleviate security concerns of potential participants. Non-probability sampling strategies were used, and we notably did not capture other professional cadres beyond doctors and nurses. The use of the English version of the survey in a population of professionals who speak English as a second language may also influence our findings, though it supports our ability to compare findings with Ayed et al [41]. While the ProQOL-H instrument offers one account of the psychological toll on health workers in harsh circumstances, the genocide is unique in its quality and severity, and qualitative work with survivors is needed to further characterize their experiences and reconcile them with culture-specific idioms of distress and culture-specific coping strategies.

Despite these limitations, we have documented some of the psychosocial dimensions of the harm done to health workers in Gaza, all of whom form a protected professional cadre under international humanitarian law, and as such should have been safeguarded. The well-being of health workers is critical to health system functioning [34,59], and the dehumanization of Palestinians enables genocide, apartheid, and the occupation [60]. Attention to the victims of this massacre and other atrocities committed by Israel in Gaza is needed to afford due recognition to the ongoing violence, and the full extent of what our Palestinian colleagues and their patients have endured. Palestinian-led research on the experiences of Palestinian health workers must serve as evidence in ongoing legal proceedings and accountability processes. Such epistemic work serves to reject the "death worlds" of Israeli necropolitics, and to ensure that the experiences of Palestinians are understood beyond a tally of brutalized or dead bodies [42].

The International Court of Justice has determined that recent events in Palestine plausibly constitute genocide. Regardless of the ultimate court ruling, this study responds to a call by Marie et al to understand sumud and other forms of anti-colonial resistance that reflect poor and dispossessed people in their particular contexts [61]. These reports help us to better understand the full experience of our Palestinian colleagues. These findings must be situated within the broader field of genocide studies, where the assessment of psychological trauma in Palestine can be interpreted with an understanding of colonial strategies of elimination [1,62]. Our study expands upon empirical work on the genocide by Hamamra et al, who conducted a qualitative study of trauma narratives and coping among 30 displaced Palestinian civilians in Rafah, interpreting their results through the lens of genocide as a distinct traumatic exposure [63]. Culturally relevant, mixed methods follow up to this body of work is needed to understand the distress of survivors and their responses to it, informing interventions which can support their recovery and the rebuilding of Gaza's healthcare system. This will only be possible when the genocide is brought to a permanent end, along with the conditions which have enabled it.

## Supporting information

**S1 Text. Survey.**
(DOCX)

**S2 Text. Open-ended Responses.**
(DOCX)

**S1 Table. Individual ProQOL-Health Response Counts, n = 35.**
(DOCX)

**S2 Table. Stratified median (IQR) ProQOL-H scores by gender, profession, most recent workplace governorate, and type of facility lived in.**
(DOCX)

## Acknowledgments

The authors would like to recognize Dr. Rola Shaheen for her steadfastness in supporting and mentoring multiple researchers on our team. This work would not be possible without her commitment. We also thank Dr. Marc LaRochelle for ethical oversight.

## Author contributions

**Conceptualization:** Mohanad S. A. Qudaih, Dalia T. A. Wehedi, Rasha Khoury, Lisa Matos, Karim Sariahmed.

**Data curation:** Mohanad S. A. Qudaih, Hamza A. Abu Daqqa, Omar R. AlNajjar, Dalia T. A. Wehedi, Karim Sariahmed.

**Formal analysis:** Mohanad S. A. Qudaih, Karim Sariahmed.

**Investigation:** Mohanad S. A. Qudaih, Hamza A. Abu Daqqa, Omar R. AlNajjar, Dalia T. A. Wehedi, Karim Sariahmed.

**Methodology:** Mohanad S. A. Qudaih, Hamza A. Abu Daqqa, Omar R. AlNajjar, Dalia T. A. Wehedi, Rasha Khoury, Benjamin Bouquet, Lisa Matos, Karim Sariahmed.

**Project administration:** Mohanad S. A. Qudaih, Karim Sariahmed.

**Resources:** Karim Sariahmed.

**Software:** Karim Sariahmed.

**Supervision:** Karim Sariahmed.

**Writing – original draft:** Mohanad S. A. Qudaih, Karim Sariahmed.

**Writing – review & editing:** Mohanad S. A. Qudaih, Hamza A. Abu Daqqa, Omar R. AlNajjar, Dalia T. A. Wehedi, Rasha Khoury, Benjamin Bouquet, Lisa Matos, Karim Sariahmed.

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
