## [Decision Letter · Decision Letter 0]

5 Feb 2025

PGPH-D-24-02208

Displacement, Personal Loss, and Psychological Strain Among Physicians and Nurses in Gaza during Israel’s Genocide in Palestine, 2023-2024

Dear Dr. Sariahmed,

Thank you for submitting your manuscript to PLOS Global Public Health. After careful consideration, we feel that it has merit but does not fully meet PLOS Global Public Health’s publication criteria as it currently stands. Therefore, we invite you to submit a revised version of the manuscript that addresses the points raised during the review process.

We look forward to receiving your revised manuscript.

Kind regards,

Tereza Hendl

Academic Editor

Journal Requirements:

1. Please provide an Author Summary. This should appear in your manuscript between the Abstract (if applicable) and the Introduction, and should be 150–200 words long. The aim should be to make your findings accessible to a wide audience that includes both scientists and non-scientists. Sample summaries can be found on our website under Submission Guidelines:

https://journals.plos.org/globalpublichealth/s/submission-guidelines#loc-parts-of-a-submission.

Additional Editor Comments (if provided):

Dear author,

We have received reviewer comments on the article. All of the reviewers appreciate the work that has gone into the analysis and view the article as an important paper accounting for the impact that Israel's genocide against the Palestinians has had on health professionals in Gaza. Yet, the reviewers have differed in their recommendations of minor and major revisions, hence, we have sought additional review statements. In the end, the majority recommends major revisions. The reviewers recommending major changes are making important suggestions, especially in regard to:

The need for the contextualization of the political context of the study: one reviewer points out that the authors state that Israel is a settler colonial regime, which is obvious to those who have studied this context, but sceptical or ill-informed readers would benefit from a line or two to explain this terminology and give a couple of basic pieces of evidence for it. Similarly, when the authors mention the Great March of Return. This is a global health journal, not a political science one, thus, readers will need some background.

The need for the contextualisation of the study: Have there been other studies of health workers during genocide or other mass atrocities? Are there specific threats genocide and settler colonial elimination pose to the conditions of these workers that are different from the direct and indirect violence of war (which they are also experiencing)?

The importance of emphasising the scale of destruction of the healthcare system and the implications of this, including the difficulty of accessing medication and other vital supplies, which has implications on the work of health professionals

The importance of contextualising the psychological strain among health providers in the wider context of siege, war, and violence

The need to contextualise debates on PTSD and go beyond debates on 'work-life balance', given the severity of the conditions

The need to provide more context on ProQOL-Health and how it relates to contexts of genocide

Concerns about de- and re-humanisation (who dehumanises Palestinian health workers and why do they need to be re-humanised?)

The need to capture the scale of the impact on healthcare workers: the toll of Israeli attack on healthcare workers - and that the death toll and destruction of the health sector has been an intentional aim of the Israeli military strategy; plus the fact that those who have survived so far have likely survived multiple wars on Gaza; some were harmed or kidnapped in the ongoing genocide and still go to work...; the need to extend on the displacement angle of the workers as well as the severity of conditions they work under, including lack of access to medication etc and the implications of this for their work and perceptions of it; that they suffer collective not only individual trauma - which all bears on the debate on impact as well as resilience and coping mechanisms

The need for greater clarity in the conceptualisation of sumud: in particular, the need to avoid conflation of sumud with compassion and the need to capture collective trauma and sumud as well as the relation of sumud and moral distress

The need for a lesser focus on author assumptions and more on the findings of the study, including, in particular: "There is a long discussion of sumud and resilience in the discussion that, while interesting, can not be elucidated from the data collected in the study. Perhaps this context would fit better in the introduction to give context as to why certain outcomes would be hypothesized."

More specifically: "Can your findings tell us more about sumud of health workers in particular? Do you find that your findings align more with Meari (reconciling victimhood and resistance) or Wick (a coping mechanism for health workers) or do they, in the context of a unique experience (a war of elimination, one of the aims of which is to dismantle the health sector as a lifeline for the Palestinian people), open up a potentially novel interpretation of sumud?"

Relatedly, further definition of terms is needed: one reviewer states that they had to look up “compassion satisfaction”, which shouldn’t be the case and the term needs to be defined in text. More importantly, lack of clarity inhibited their understanding of the findings: "Is the central claim that sumud is protective against the kind of serious moral distress that might be expected of healthworkers in a genocide? If so, say so directly! But also say what kind of expectations you had, and what those expectations are based upon. Have similar studies been done in other conflict situations, and the moral distress was greater? I’m a bit lost as to what is being concluded. At another point, the authors mention that PTSD is not appropriate in relation to the violence experienced by Palestinian people, but they don’t explain it. More generally, the authors need to slow down and give more definitions, more explanations, more real-world examples, and more references."

Methods: the need to clarify whether the sample includes mainly Palestinian health workers, or anyone who worked/volunteered in the Gaza health sector before and sometime after October 7th

the need to clarify on reasons why some participants refused participation in the study (if known)

the need to explore the challenges and limitations regarding the conduct of the survey in English and not Arabic, given the topic of trauma and related research on trauma-related methodologies

sample: demographic characteristics and the role of gender

The need to restructure the discussion section: one reviewer points out that the flow needs to be improved. "The discussion section in general is quite unclear and needs to be almost entirely rewritten. It starts with a paragraph that is a page long and that is not so much discussion of the data (as it should be) as a political update on the situation. I don’t object to this information being included, but first you should discuss your data! This is the section in which you need to carefully describe your expectations regarding compassion fatigue and carefully explain why you think they aren’t met here."

Please also have a look at reviewer's minor comments which all raise points regarding flow of sentences etc, which aim to contribute to the improvement of the paper, that is overall perceived by the reviewers as an important contribution to debates and empirical evidence on the impact of Israel's genocide on Palestinians in Gaza on healthcare workers.

Reviewers' comments:

Reviewer's Responses to Questions

**Comments to the Author**

1. Does this manuscript meet PLOS Global Public Health’s publication criteria?

Reviewer #1: Yes

Reviewer #2: Yes

Reviewer #3: Yes

Reviewer #4: Yes

Reviewer #5: Yes

2. Has the statistical analysis been performed appropriately and rigorously?

Reviewer #1: N/A

Reviewer #2: Yes

Reviewer #3: I don't know

Reviewer #4: No

Reviewer #5: I don't know

3. Have the authors made all data underlying the findings in their manuscript fully available (please refer to the Data Availability Statement at the start of the manuscript PDF file)?

Reviewer #1: Yes

Reviewer #2: Yes

Reviewer #3: Yes

Reviewer #4: Yes

Reviewer #5: Yes

4. Is the manuscript presented in an intelligible fashion and written in standard English?

Reviewer #1: Yes

Reviewer #2: Yes

Reviewer #3: Yes

Reviewer #4: Yes

Reviewer #5: Yes

Reviewer #1: Dear Authors,

Thank you for conducting and writing this important and timely research.

In this paper, the authors try to provide an insight into the psyche of healthcare workers working during a genocide while witnessing horrors and fearing for their lives and their loved ones.

The paper is well written, and the authors use accurate and important framework of necropolitics and settler colonialism.

I am very impressed you were able to conduct such a study in those conditions considering the limited access to health care workers and their constant displacement.

Still, some revisions are required to strengthen the arguments of the article. Kindly see comments below:

1. Lines 194-197 are not clear and can be omitted

2. Table 1 - please add a column of percentages near the number of participants and a row for “total”.

3. Table 2 – please remove hospitals that have 0 workers in both columns like Al Dorrah and Al Rantisi.

4. This is a useful article for the discussion of “Sumod”

https://thefunambulist.net/magazine/redefining-our-terms/nakba-sumud-intifada-a-personal-lexicon-of-palestinian-loss-and-resistance

5. The discussion needs to be fleshed out. One of the main findings is that the available questionnaires cannot measure the magnitude of the experience of health workers in Gaza and their comping mechanisms. I understand that it is impossible to conduct larger study at the moment but the authors should point out to the need of conducting open interviews after the genocide is over and synthesize new tools to explore the mental effects of being and treating genocide victims such as grief, bereavement, shock, collective trauma and Sumoud. See for example:

https://onlinelibrary.wiley.com/doi/abs/10.1002/smi.1429?casa_token=Fzu6mzNU5pkAAAAA%3ALHS6mCTUXd3K5x34mdRydK3BNMYmY9OgVmeSah0FELVGwQ7NdIE9IY6ROFPEBWKozNPr74x_7isU

6. The authors should discuss that these health workers have probably survived several Israeli wars on Gaza, a fact that perhaps strengthen their coping mechanism. The authors are also expected to discuss briefly the previous studies of Gaza healthcare workers during previous Israeli wars, see the multiple works of Abdelhamid Afana, Guido Veronese and others, they are certainly more relevant than some of the studies mentioned about healthcare workers coping with COVID. see for example:

https://journals.sagepub.com/doi/abs/10.1177/1359105318785697

https://psycnet.apa.org/buy/2020-54569-001

https://www.sciencedirect.com/science/article/pii/S0883941717300316

https://academic.oup.com/bjsw/article-abstract/43/4/651/1643017

7. The mentioned healtcare workers, as everyone in Gaza, are also going through a collective trauma that cannot be measured by simply examining multiple individual experiences. It is worth bringing this aspect and the need for further exploration of collective trauma and collective Sumud, see for example:

https://www.sciencedirect.com/science/article/pii/S2666560324000264

8. It is also worth adding to the discussion that the genocide has targeted many hospital and many healthcare workers were killed, abducted and tortured. The fact that these healthcare workers still go to work every day is an act of heroism and resistance that should be saluted.

Reviewer #2: This is an extremely important paper. Thanks and recognition must in particular be given to the Gaza-based authors and participants who have put in the time and effort to create knowledge despite being subjected to a war of elimination in the ongoing genocide.

The methods are appropriate considering the current circumstances, which lead to the limitations recognised by the authors. I suggest some small modifications that would strengthen the paper and help guide future work on the topic, whether by the same authors or others who wish to conduct work on the same topic.

General note

There is obviously a lot to cover in a small amount of space, but considering that the ProQOL includes questions related to the work itself and the work environment in general, it would be important to present (either in the background or the discussion) the potential effect of the siege and targeting of facilities. These have led to serious challenges in carrying out the necessary health work due to destruction or shortage of equipment, disposables, medications, etc. This may affect health workers’ perception of their own work and its limitations (for example, it could on the one hand make them less satisfied with the work because they cannot perform their best, or it could make them more proud because of making the best of a terrible situation).

Background

Lines 79-80: because the focus of this paper is on health workers, it would be prudent to include additional detail on the toll of Israeli attacks on healthcare workers. For example, you could name the most recent data on health workers killed (total of 986 in the August 2024 report by the Palestinian Ministry of Health) or incarcerated (310 according to the September 25, 2024 update by the Palestinian Government Media Office). You could also include some detail of how and where health workers have been targeted, for example through precision killings whether at work or at home, abductions from workplaces or as they took designated “safe routes”, or as they travelled as part of convoys organised by the WHO and/or ICRC.

In addition to this, you can also point to the conclusion reached by many who experienced and/or witnessed the genocide that the destruction of the health sector was a key aim of the Israeli military strategy.

Methods

The use of an online survey presents limitations, as you discussed, because of the destruction of infrastructure leading to connectivity problems. Did you consider a paper-based survey? Why/why not?

Lines 149-154: please explain why you chose to include only physicians and nurses at this stage (especially as you note this as a limitation in the discussion)

Line 194: “Palestinian members of our team and their” – I think there is a word missing after “their”

Lines 194-195: it would be important to specify what exactly you found inappropriate about the questions about health worker treatment by managers in the Arabic version of the ProQOL tool. This would allow for a more open debate about how to modify the Arabic version of the tool in the future.

Discussion

Lines 309-310: since several of the authors are health workers from Gaza themselves, this could provide an opportunity to include additional detail on what you interpret to be the potential “cultural differences in the perception of trauma” and “differences in strengths and protective factors”, especially since Gazan health workers are likely to have witnessed previous assaults on Gaza and the pandemic, before the ongoing genocide.

Your interpretation of the findings through the lens of sumud is profoundly important and hopefully the start of further debate. Can your findings tell us more about sumud of health workers in particular? Do you find that your findings align more with Meari (reconciling victimhood and resistance) or Wick (a coping mechanism for health workers) or do they, in the context of a unique experience (a war of elimination, one of the aims of which is to dismantle the health sector as a lifeline for the Palestinian people), open up a potentially novel interpretation of sumud? You don’t have to have the answer, but there is an opportunity for a valuable contribution to the debate (or some new questions), especially when considering, for example, the defiance of Palestinian health workers in the face of threats by the Israeli army and instructions to evacuate, and their insistence to continue operating hospitals as a patriotic and ethical duty.

Although this is not the main aim of the paper, can you speak to the appropriateness of the ProQOL tool itself for the current context of the genocide? Were there areas that needed to be included that were missing? Or others that were included that were not appropriate for the context?

Based on your findings and interpretation, are you able to suggest future work on this topic, whether that you are planning as a team or that you suggest others to conduct?

Reviewer #3: Thank you for writing this important piece.

Abstract:

The abstract is well written and clear. I would suggest to also refer to the ICJ and its claim that what we are witnessing is genocide.

Main document:

It would be helpful to start with a brief introduction that outlines the objective of the study, justifies its importance, outlines the research gap, and lists the research questions with which the authors seek to fill the gap. An outline of the article and how it contributes to research and practice would also be helpful. This would help the reader to understand the relevance of the following sections.

Background:

Overall, the background section could be better structured. Right now it is not always easy to follow. My suggestion is:

(a) Background on the current genocidal war on Gaza and connected to this the number of people killed and injured, neighbourhoods flattened, and vital infrastructure destroyed. (Here you can refer immediately to Albanese’s report and the ICJ and other documentation)

(b) Focus on the health system (what used to be there); the almost total destruction of the health infrastructure; and the conditions under which health professionals have to work

(c) The impact of working under such conditions (while living through the genocide themselves) on health workers’ health and mental health as far as we know it

(d) Contextualisation of their experiences in relation of previous onslaughts and what we know about their health and mental health during these. The expected cumulative effect can be mentioned which brings it back to the present.

(e) Final note of why understanding health workers mental health is so important

Other comments on background:

It would be important to state that the health system has been completely destroyed and brough to its knees. More information about the destruction of infrastructure and the difficulty of accessing medication and other vital supplies would provide important contextual insight to better understand the almost impossible situation in which health workers need to carry out their work.

The psychological strain among health providers needs to be understood in the wider context of siege, war, and violence. Would it be useful to also refer to other recent onslaughts in 2008/2009, 2012, 2014, and 2021 and then of course the Great March of Return in 2018 (you mention the latter) which caused many injuries and disabilities especially among young people. Health professionals will have seen horrific injuries and other war related health problems before the genocide which leave their psychological marks.

Methods:

The methods section seems to suggest that you didn’t only include Palestinian health workers, but anyone who worked/volunteered in the Gaza health sector before and sometime after October 7th. I think this needs to be made much clearer from the start (i.e., objective and research questions (which are absent). It would also need to feature in the background more strongly. The background needs to highlight that Gaza’s health system is supported by international health workers/volunteers with relevant numerical and demographical information provided.

Results:

These are important and well reported.

While you moved qualitative quotes in the appendix as there were too few for thematic analysis, I wonder if they could be included in relevant sections as illustrative examples of how participants reflected on what the numbers reveal. You can note the lack of methodological rigour here in the methods section – I just find that these quotes bring the numbers to life and form part of your findings.

Discussion:

Much of the information here reads like it belongs in the background section. It would have been useful for readers to read about the destruction of hospitals and the mass graves that were uncovered in them beforehand.

While the reference to sumud is apt, it did not come up in the survey. Conflating sumud with compassion might not be entirely appropriate. It would have been good for the survey to reflect more locally relevant expressions of distress and strengths. This, in my view, is another key limitation: i.e., Western conceptions were inquired into while the interpretation tries to make them locally/culturally relevant without having the necessary data. Considering this, the section on sumud is too detailed and long. Instead, it would have been useful to know how these findings reflect experiences of health workers in other war affected settings. What is particular about this situation, what similarities exist?

Finally, I would have liked to know what practical lessons the research inspires. What should be done to support health providers in the short-, medium-, and long-term?

Reviewer #4: Important and meaningful context in para 2, but seems a bit disjointed, especially the reference to food and water deprivation (which seems unrelated to the issue of health care workers—perhaps make more clearly the point that these workers are suffering from the same deprivation and starvation). It appears that the thesis of the paragraph is in the last sentence, so I would rearrange this content by streamlining this and the next paragraph (starting with line 91).

The context of the Albanese report should lead the following paragraph, as it offers the framing for the rest of the piece. I recognize that genocide is the context in which the workers being studied are living and working under, but the focus of the introduction should be more heavily focused on how the conditions of genocide manifest in these worker’s living and working conditions, and less so on the wider context itself, as to reflect the study.

Should there be earlier reference to the ICJ case, and how Albanese’s report offers evidence for the legal definition of genocide? This is not clear in lines 103-107.

The framework of settler-colonialism is brought in early in para 4, but requires more explanation for a general public health audience. It seems “apartheid regime” may be more relevant considering the rest of the content in the sentence— otherwise, more explicitly make the connection between genocide and settler colonial elimination for the reader.

The paragraph starting on line 111 seems to be unrelated to the ultimate thesis of the paper. I think you could go directly to line 120 and then into the next para.

Line 126- some context for Great March of Return needed for those who are not familiar.

A stronger case needs to be made in the paragraph leading up to the research statement starting on line 133. I also don’t think enough has been done to bridge the gap between the statement that PTSD and other such measures have been widely critiqued in Palestine and other crisis-affected settings (long before the GMR) and the justification for using similar measures in this study.

The recruitment challenges are completely understandable, but can you clarify whether potential participants told study authors the reasons for not participating, or whether these were logical assumptions made by study authors (because this is an important finding in and of itself and is an important comment to the difficulty of doing these work in such settings).

Interesting that the survey was conducted in English— there is an abundance of literature that demonstrates that part of capturing trauma could include the ability for participants to relay their experiences in their native language, in which different words and concepts may be interpreted differently. Even though “English is spoken by the majority of physicians and nurses in Palestine” as the authors state, could you say more about this decision?

The sentence starting on line 194 seems missing a word or two.

To be clear, were the 79 individuals approached those approached by the seed participants as described earlier? How was it shared with researchers how many were approached in total?

Do the recruitment efforts skewing male parallel the overall gender disparity in medical workers in Gaza/Palestine more broadly?

Table 2 is very interesting, tracing the displacement of the workers. Would be very useful to see this translated to a line or bar graph (perhaps just the governorates as opposed to the individual hospitals).

Concepts like “work-life balance” seem insufficient in the face of what these workers are experiencing. Could you describe how well the ProQOL-Health mapped onto capturing meaning in the lived experience of the participants?

More detail about the ProQOL-Health is needed. Has it been used in other settings of war/genocide? In the discussion, a few studies are addressed, but it’s unclear that the work of nurses in Jordan during COVID bears much relation to the experiences of health workers in Gaza today.

This descriptive analysis would be made significantly more robust with inferential statistics. I also think it would be interesting to see whether any of the demographic characteristics had any bearing on the results, for example by gender, by the type of facility in which they are staying, or by profession. Does that affect their level of burnout or perceived support, for example? The discussion would be greatly strengthened if it depended more on the results presented from the study and explored them.

Disappointing that there were insufficient qualitative responses, as I think this could make the study more robust.

While I wholeheartedly agree with the second half of the sentence (line 294), I’m not sure how to interpret the beginning : “Capturing the experiences of Palestinian health workers forms part of a wider process of re-humanisation”—to whom are these workers re-humanizing themselves?

The relationship between sumud and moral distress is unclear to me. I assume these workers perceive they are acting with high ethics, but are just limited by circumstance. Thus, it is unclear why they would be assumed to have high moral distress. I have similar questions about the compassion satisfaction measure. I would assume, again, that these workers do feel good about helping their people despite the conditions, and in fact that is what keeps them able to work. Why is the assumption that it would be low? I think it would be useful to define these constructs earlier in the paper and the factors that have been shown to affect them.

I am not sure the framing of “protected by resilience” is sufficiently explained in line 330.

Overall, this paper offers a unique insight into healthcare workers that have the unprecedented task of attempting to deliver care, and survive, while surrounded by countless threats. The ability to conduct this study at all at this time shows remarkable commitment, and the perspectives of these workers in such a time are invaluable, including as evidence as is noted in the discussion.

However, the introduction needs to be brought together into a cohesive background and justification for the study at hand, including discussing the experiences of these healthcare workers (there are some examples of these, but it needs to be presented more systemically). Have there been other studies of health workers during genocide or other mass atrocities? Are there specific threats genocide and settler colonial elimination pose to the conditions of these workers that are different from the direct and indirect violence of war (which they are also experiencing)?

Despite the obvious and acceptable limitations of conducting such a study, the data analysis could be more rigorous. The demographic information is not as useful if it is not tied to specific outcomes of the study. Some details of the methodology also need to be more clear, not just for the benefit of this study but for the benefit of other researchers who wish to conduct similar studies under such conditions across contexts. I also think use of the ProQOL-Health tool needs to be better justified, as there appear to be some aspects of it that are less relevant in this setting, and it is not being coupled with qualitative analysis to help describe some of the outcomes.

I think the displacement angle of the workers is an interesting finding, and should be described and discussed more.

The discussion should focus more on explaining the results found, and explaining some of the assumptions seemingly made by the authors about what was found in the study. There is a long discussion of sumud and resilience in the discussion that, while interesting, can not be elucidated from the data collected in the study. Perhaps this context would fit better in the introduction to give context as to why certain outcomes would be hypothesized.

Reviewer #5: Thanks for the opportunity to review this paper, which studies the impact of Israel’s genocide in Gaza on healthcare workers in settings across Gaza. The authors have conducted the ProQl-Health survey, and have studied the responses of 56 participants. This is a valuable piece of work in that it provides important information about the effects of the genocide which will be of use to scholars of health and genocide, as well as policy-makers. It is not publishable in its current form, but I am supportive of its eventual publication. However, I have some serious reservations that I think should be addressed in a revised version of this work.

1. There isn’t enough exposition on the political context of this article. While I agree with the ideas expressed in this work, not everyone will, and more explanation is needed. The authors state that Israel is a settler colonial regime, which is obviously true to those of us who have studied this context, but sceptical or ill-informed readers would benefit from a line or two to explain this terminology and give a couple of basic pieces of evidence for it. Similarly, when the authors mention the Great March of Return. This is a global health journal, not a political science one. Readers will need some background.

2. There also isn’t enough exposition on the psychological details of the article. This is a much more serious concern, because it’s currently preventing me from understanding the authors’ findings. I had to look up “compassion satisfaction” (I was chosen to review this article based on my knowledge of Palestine, not psychology), which shouldn’t be the case for technical words. But more importantly, I don’t understand the authors’ findings. Is the central claim that sumud is protective against the kind of serious moral distress that might be expected of healthworkers in a genocide? If so, say so directly! But also say what kind of expectations you had, and what those expectations are based upon. Have similar studies been done in other conflict situations, and the moral distress was greater? I’m a bit lost as to what is being concluded. At another point, the authors mention that PTSD is not appropriate in relation to the violence experienced by Palestinian people, but they don’t explain it. More generally, the authors need to slow down and give more definitions, more explanations, more real-world examples, and more references.

3. The discussion section in general is quite unclear and needs to be almost entirely rewritten. It starts with a paragraph that is a page long and that is not so much discussion of the data (as it should be) as a political update on the situation. I don’t object to this information being included, but first you should discuss your data! This is the section in which you need to carefully describe your expectations regarding compassion fatigue and carefully explain why you think they aren’t met here.

**Do you want your identity to be public for this peer review?** For information about this choice, including consent withdrawal, please see our Privacy Policy

Reviewer #1: No

Reviewer #2: **Yes: ** Layth Hanbali

Reviewer #3: No

Reviewer #4: No

Reviewer #5: No

---

## [Decision Letter · Decision Letter 1]

14 May 2025

PGPH-D-24-02208R1

Displacement, Personal Loss, and Psychological Strain Among Physicians and Nurses in Gaza during Israel’s Genocide in Palestine, 2023-2024

Dear Dr. Sariahmed,

Thank you for submitting your manuscript to PLOS Global Public Health. After careful consideration, we feel that it has merit but does not fully meet PLOS Global Public Health’s publication criteria as it currently stands. Therefore, we invite you to submit a revised version of the manuscript that addresses the points raised during the review process.

We look forward to receiving your revised manuscript.

Kind regards,

Tereza Hendl

Academic Editor

Journal Requirements:

Additional Editor Comments (if provided):

Dear authors,

the reviewers have appreciated the revisions and assessed them positively. They still request a few minor changes. Please address these comments. Your paper is recommended for publication, pending on the few minor changes that will streamline the text and make it more accessible, improving the flow of the argumentation line and analysis.

Reviewers' comments:

Reviewer's Responses to Questions

**Comments to the Author**

Reviewer #2: All comments have been addressed

Reviewer #3: No response

publication criteria?

Reviewer #2: Yes

Reviewer #3: Yes

3. Has the statistical analysis been performed appropriately and rigorously?

Reviewer #2: Yes

Reviewer #3: I don't know

4. Have the authors made all data underlying the findings in their manuscript fully available (please refer to the Data Availability Statement at the start of the manuscript PDF file)?

Reviewer #2: Yes

Reviewer #3: Yes

5. Is the manuscript presented in an intelligible fashion and written in standard English?

Reviewer #2: Yes

Reviewer #3: Yes

Reviewer #2: Thank you so much for your work on this. It's really an important and excellent paper, and the revisions you have made greatly improve it. Just one small comment:

Lines 118-127: in addition to the challenges to PTSD frameworks that you identified, it’s also important to mention the debate about whether PTSD can even be applied in the Palestinian (especially Gazan) context, due to the fact the trauma has not ended. See Samah Jabr for example.

Reviewer #3: Thank you for sharing the revised version of this important manuscript. I believe it is nearly there. However, the background and discussion sections would benefit from further streamlining and greater clarity regarding the study’s objective, research questions, and relevant context. As it stands, the reader may struggle to grasp the focus of the article at the outset. That said, the methods section has improved, and the results are both highly informative and important

Abstract:

It would have been helpful to know who the participants were—specifically, that they were physicians and nurses from Gaza and international staff working in Gaza.

Background:

Overall, the background section needs to be more streamlined toward the objective of the study which is also not clearly outlined. What is the objective, what are the research questions, and what background information is relevant to understand the results?

Intro of background section:

The background information is as such good, but it is difficult to know how it relates to the study objective. Is there any way that the authors can lead with their objective and then outline the context in which the study is carried out – one of illegal occupation, apartheid, and genocide. Also, in relation to the genocide, the health-related consequences could have been highlighted in a more integrated way. At this point, the readers doesn’t learn anything about the disastrous health situation that physicians and nurses are dealing with. I find this an oversight.

It would have been useful to end the section with a clear objective and research questions that guide the piece (I am not convinced that the piece is making a theoretical contribution; it is making an important empirical one).

Targeting of Health Workers and Health Infrastructure in Gaza:

P7: The text would benefit from a more structured flow. It currently jumps between affected people, infrastructure, and patterns of attack. I suggest starting with the impact on people, then addressing infrastructure, and finally discussing attack patterns—without shifting back and forth between these topics.

Psychological Constructs for Health Worker Distress and Well-being:

I would have moved this section following the section ‘Studying the Psychological Effects of Genocide’ – like this it interrupts the flow from understanding how the horrendous situation in Gaza affects health workers. This can then be followed by a section about how the international literature has tried to make sense of this more generally by introducing the various relevant concepts.

Studying the Psychological Effects of Genocide:

The section needs to be more focussed on the psychological effects of genocide among healthcare workers. The destruction of the social and cultural fabric could have been mentioned in the background above rather than here. That is, after a section that outlined the targeting of healthcare workers, I would have expected a focus on the evidenced and/or expected psychological health consequences. The authors could have engaged with health reports to explore how health workers express their distress.

Methods:

Thank you for improving this section

Results:

Results are presented clearly and are informative.

A stylistic note: P16/line292: “Additionally, 41 health workers…” using ‘additionally’ makes the experiences of personal loss sound slightly like an afterthought. I would say: “Personal losses were experienced by 41 health workers due to Israeli military violence (Table 3)…”

Discussion:

The discussion summarises the key findings well.

I still struggle somewhat with the extrapolation from the literature on idioms of distress and steadfastness to explain perceived support and levels of moral distress. One aspect that seems overlooked, for example, is the role of adrenaline in keeping people going under even the most horrendous circumstances. If the authors wish to keep the text as it stands, it would benefit from further streamlining and clarification how experiences of trauma, mental distress, and steadfastness might affect the ways in which health workers regarded their level of distress and support received.

**Do you want your identity to be public for this peer review?** For information about this choice, including consent withdrawal, please see our Privacy Policy

Reviewer #2: No

Reviewer #3: No

---

## [Editor Report · Decision Letter 2]

20 Jul 2025

PGPH-D-24-02208R2

Displacement, Personal Loss, and Psychological Strain Among Physicians and Nurses in Gaza during Israel’s Genocide in Palestine, 2023-2024

Dear Dr. Sariahmed,

Thank you for submitting your manuscript to PLOS Global Public Health. After careful consideration, we feel that it has merit but does not fully meet PLOS Global Public Health’s publication criteria as it currently stands. Therefore, we invite you to submit a revised version of the manuscript that addresses the points raised during the review process.

Please submit your revised manuscript by . If you will need more time than this to complete your revisions, please reply to this message or contact the journal office at globalpubhealth@plos.org. Please include the following items when submitting your revised manuscript:

We look forward to receiving your revised manuscript.

Kind regards,

Tereza Hendl

Academic Editor

Journal Requirements:

Additional Editor Comments (if provided):

Dear authors,

Thank you very much for the revisions you have made to the paper, which have strengthened it and ensured that it is almost ready for publication. Thank you also for your patience as I have consulted with the Senior Editorial team at PLOS regarding your paper. We are ready to move forward, but first there is a request regarding a change to the title of the manuscript, a slight modification to the abstract of the paper, and an additional suggested reference.

Regarding the title, PLOS has found that general and social media coverage tends to include the title of articles only; researchers often read only title and abstract; as such, the important context and explanation around use of the term ‘genocide’ provided in the body text will likely not be seen by most readers. The presumptive lack of that context and explanation is likely to trigger strong negative reactions (perhaps more in some media contexts than in others), including accusations of bias/partisanship. If accusations of bias/partisanship dominate the media coverage of/reader reaction to the article, the actual content of the article—and much of the important impact it could have—can likely be overshadowed.

Thus, a suggestion has been made regarding the following alternative phrases to be included in the title in place of the word "genocide":

Israel's war on Gaza

Israel's onslaught on Gaza

Israel's assault on Gaza

Israel's ongoing offensive in Gaza

Israel's siege in Gaza

We'd also recommend that you add a sentence to the abstract that summarizes this excerpt from the introduction: ‘“We use the word “genocide” to refer to this period of accelerated violence and displacement in Gaza, following preliminary ICJ findings and subsequent detailed reporting by UN Special Rapporteur Albanese.(4,5) It is also consistent with more recent reports of the United Nations Special Committee…’ so as to provide additional context in the abstract itself. This is because readers and the media often only read the title and the abstract of research papers and might miss the careful justification you include around the choice of this framework.

Finally, we recommend that you cite a recent peer-reviewed paper that uses genocide as framework as a precedent to further substantiate the concepts they use (Hamamra, B., Mahamid, F., Bdier, D., & Atiya, M. (2025). War-related trauma in narratives of Gazans: challenges, difficulties and survival coping. Global mental health (Cambridge, England), 12, e34. https://doi.org/10.1017/gmh.2025.23)

We hope that the reasoning behind these recommendations is understandable and that we will be able to find an agreeable way forward. PLOS is eager to share the results of this very important research and wants to ensure that it can be done in a way that upholds the contributions while ensuring the work is received and understood in its rigorous research and crucial human context.

Kind regards,

Tereza Hendl and the PLOS editorial team
---

## [Editor Report · Decision Letter 3]

4 Aug 2025

Displacement, Personal Loss, and Psychological Strain Among Physicians and Nurses Working in Gaza, 2023-2024

PGPH-D-24-02208R3

Dear Dr Sariahmed,

We are pleased to inform you that your manuscript 'Displacement, Personal Loss, and Psychological Strain Among Physicians and Nurses Working in Gaza, 2023-2024' has been provisionally accepted for publication in PLOS Global Public Health.

Best regards,

Tereza Hendl

Academic Editor

Dear authors, thank you for integrating the suggested revisions and congratulations on the acceptance of your important work for publication in PLOS. May your empirically substantiated paper inform global health debates and policy for the better.

Kind regards,

Tereza Hendl